# FINITE SCALAR QUANTIZATION: VQ-VAE MADE SIMPLE

**Fabian Mentzer[1], David Minnen[1], Eirikur Agustsson[1], Michael Tschannen[2,∘]**
[1]Google Research [2]Google DeepMind

## ABSTRACT

We propose to replace vector quantization (VQ) in the latent representation of VQ-VAEs with a simple scheme termed finite scalar quantization (FSQ), where we project the VAE representation down to a few dimensions (typically less than 10). Each dimension is quantized to a small set of fixed values, leading to an (implicit) codebook given by the product of these sets. By appropriately choosing the number of dimensions and values each dimension can take, we obtain the same codebook size as in VQ. On top of such discrete representations, we can train the same models that have been trained on VQ-VAE representations. For example, autoregressive and masked transformer models for image generation, multimodal generation, and dense prediction computer vision tasks. Concretely, we employ FSQ with MaskGIT for image generation, and with UViM for depth estimation, colorization, and panoptic segmentation. Despite the much simpler design of FSQ, we obtain competitive performance in all these tasks. We emphasize that FSQ does not suffer from codebook collapse and does not need the complex machinery employed in VQ (commitment losses, codebook reseeding, code splitting, entropy penalties, etc.) to learn expressive discrete representations. **Colab on GitHub**.

## 1 INTRODUCTION

Vector quantization (VQ), initially introduced by Gray (1984), has recently seen a renaissance in the context of learning discrete representations with neural networks. Spurred by the success of VQ-VAE (Van Den Oord et al., 2017), Esser et al. (2020) and Villegas et al. (2022) showed that training an autoregressive transformer on the representations of a VQ-VAE trained with a GAN loss enables powerful image and video generation models, respectively. At the same time, VQ has become popular component in image (Bao et al., 2021; Li et al., 2023) and audio (Baevski et al., 2019) representation learning, and is a promising building block for the next generation of multimodal large language models (Aghajanyan et al., 2022; Kim et al., 2023; Aghajanyan et al., 2023).

When training VQ-VAE, the goal is to learn a codebook $\mathcal{C}$ whose elements induce a compressed, semantic representation of the input data (typically images). In the forward pass, an image $x$ is encoded into a representation $z$ (typically a sequence of feature vectors), and each vector in $z$ *quantized* to (*i.e.*, replaced with) the closest vector in $\mathcal{C}$. The quantization operation is not differentiable. When training a VAE with VQ in the latent representation, Van Den Oord et al. (2017) use the straight-through estimator (STE) (Bengio et al., 2013), copying the gradients from the decoder input to the encoder output, resulting in gradients to the encoder. Since this still does not produce gradients for the codebook vectors, they further introduce two auxiliary losses to pull the codeword vectors towards the (unquantized) representation vectors and vice-versa.

The above formulation is challenging to optimize, and leads to the well-documented problem of underutilized codebooks (Łańcucki et al., 2020; Takida et al., 2022; Dhariwal et al., 2020; Huh et al., 2023): as the size of $\mathcal{C}$ is increased, many codewords will be unused. Subsequent works aimed to improve this with various tricks such as reinitializing the entire codebook or some codewords Dhariwal et al. (2020); Łańcucki et al. (2020), stochastic formulations Takida et al. (2022), *etc.* (see Sec. 2).

---

∘Significant technical contributions.

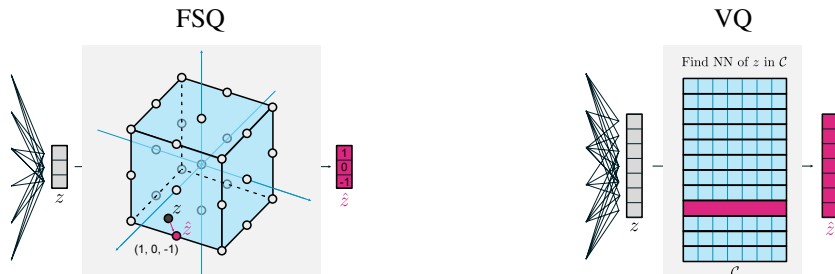

Figure 1: *FSQ (left):* the final encoder layer projects to $d$ dimensions ($d = 3$ shown). We bound each dimension of the encoder output $z$ to $L$ values ($L = 3$ shown), and then round to integers, resulting in the quantized $\hat{z}$, the nearest point in this hypercube. *VQ (right)*: The final encoder layer projects to $d$ dimensions ($d = 7$ shown, as $d$ is typically much larger for VQ). The resulting vector $z$ is replaced with the closest vector from the codebook, $\hat{z}$, by nearest neighbor lookup.

Here, we are interested in simplifying the original VQ-VAE formulation (Van Den Oord et al., 2017) with the following goals: i) remove auxiliary losses, ii) achieve high codebook utilization by design, and iii) keep the functional setup the same to the extent that we obtain a *drop-in replacement for VQ*.

To this end, we draw inspiration from the neural compression literature, where discrete codes are typically obtained with scalar quantization, following initial work (Ballé et al., 2016; Theis et al., 2017): Each (scalar) entry in the representation $z$ is independently quantized to the nearest integer by rounding. The majority of the current compression literature uses *unbounded* scalar quantization, where the range of integers is not limited by the encoder, only by constraining the entropy of the representation. Other compression work relied on *bounding* the range of the quantizer (Mentzer et al., 2018; Tschannen et al., 2018; Agustsson et al., 2019).

We call this approach finite scalar quantization (FSQ). The important insight is that by carefully choosing how to bound each channel, we can get an *implicit* codebook of (almost) any desired size: Consider a vector $z$ with $d$ channels. If we map each entry $z_i$ to $L$ values (*e.g.*, via $z_i \mapsto \lfloor L/2 \rfloor \tanh(z_i)$ followed by rounding to integers), we obtain a quantized $\hat{z}$, where $\hat{z}$ is one of $L^d$ unique possible vectors. Fig. 1 shows FSQ for $d=3, L=3$, implying a codebook $\mathcal{C} = \{(-1, -1, -1), (-1, -1, 0), (-1, -1, 1), \ldots, (1, 1, 1)\}$, where $|\mathcal{C}| = L^d = 27$.

To get gradients through the rounding operation, we use the STE like VQ-VAE. Thus, using FSQ inside an autoencoder trained with a reconstruction loss, we get gradients to the encoder that force the model to spread the information into multiple quantization bins, as that reduces the reconstruction loss. **As a result, we obtain a quantizer that uses all codewords without any auxiliary losses.**

To the best of our knowledge, considering the (product) codebook obtained from FSQ has not been done before, neither in neural compression nor in other tasks where VQ is dominant. We aim to change this by revisiting FSQ in conjunction with powerful transformers/language models. In summary, our contributions are:

1. We show that FSQ can serve as a drop-in replacement for VQ in various architectures, for different datasets and tasks, by applying it to MaskGIT (Chang et al., 2022) for image generation, and in UViM (Kolesnikov et al., 2022) for depth estimation, colorization, and panoptic segmentation. We observe a reduction of only 0.5 - 3% in the respective metrics, and correspondingly get highly similar visual results. We emphasize that the two model families have very different designs (convolutional vs. transformer-based autoencoders, masked vs. fully autoregressive transformers, decoder-only vs. encoder-decoder transformers, etc.).

2. We analyze the trade-offs for VQ vs. FSQ, characterize the scaling behaviors w.r.t. codebook size of the two models, and analyze the representation complexity from a compression angle. We find that FSQ is able to leverage large codebooks for better reconstruction metrics, and better sample quality. The codebook usage is very high for FSQ ($\approx 100\%$ for most models), without relying on any auxiliary losses.

3. We show that the full generality of the VQ formulation gives little benefits over our simpler FSQ method (VQ is actually worse for large codebooks $\mathcal{C}$). This can be attributed to VQ being difficult to optimize, whereas FSQ can be viewed as the standard VQ formulation changed such that a)

| | VQ | FSQ |
|---|---|---|
| Quantization | $\arg\min_{c \in \mathcal{C}} \|z - c\|$ | $\text{round}(f(z))$ |
| Gradients | STE | STE |
| Aux. Losses | Commitment, codebook, entropy loss | - |
| Tricks | EMA on codebook, codebook splitting projections, . . . | - |
| Parameters | Codebook | - |

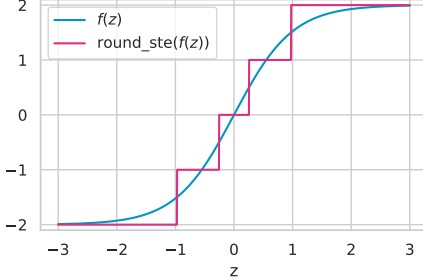

Figure 2: *Left*: VQ made simple: comparing implementation and optimization of VQ vs. FSQ. *Right*: Bounding $z$ with $f$, and rounding the output, shown for a single channel with $L = 5$.

the encoder output is bounded and b) $\mathcal{C}$ is fixed. We note that the (implicit) FSQ $\mathcal{C}$ has much smaller dimensionality vs. VQ (typically $d < 10$ for FSQ, vs. $d \geq 512$ for VQ).

## 2 RELATED WORK

**VQ-VAE and improvements** Van Den Oord et al. (2017) introduced the initial formulation in VQ-VAE, including a commitment loss and EMA for improved codebook learning. Roy et al. (2018) use soft expectation maximization (EM) to train VQ-VAE. They also report success in tuning the codebook size for the target tasks. Dhariwal et al. (2020) use VQ-VAE for audio generation. To prevent codebook collapse, they use "random restarts", where vectors are reset to encoder outputs when their usage becomes low. They also introduce a multi-scale variant of VQ. Łańcucki et al. (2020) aim to improve codebook learning by periodically reinitializing it using offline clustering algorithms. Yu et al. (2021) introduce a vision transformer (ViT) based VQ-GAN. To improve learning of the quantizer, they $l_2$-normalize all vectors and map codes to a lower dimensional space for lookup. Takida et al. (2022) propose a stochastic quantization approach to avoid codebook collapse, adding Gaussian noise to the encoder output to imitate quantization, which is annealed during training. Williams et al. (2020) also explore stochastic quantizers, in addition to a hierarchical representation. Huh et al. (2023) examines challenges in training the vanilla VQ formulation. They propose various improvements, including a re-parameterization, alternating optimization, and an improved commitment loss.

**VQ Alternatives** Residual quantization (RVQ) has been used for image (Lee et al., 2022) and audio (Zeghidour et al., 2021) generation. There, quantized codes are refined by additionally storing (quantized) residuals. In Product quantization (PQ) (El-Nouby et al., 2022), the codebook is factored into a product of smaller codebooks. In a similar spirit, there is a body of literature around reducing the number of tokens output by VQ-VAEs for more efficient inference, see, *e.g.*, Huang et al. (2023).

**Neural compression** Many works (Ballé et al., 2016; Minnen et al., 2018; Lu et al., 2019; Mentzer et al., 2020; Cheng et al., 2020) rely on unbounded scalar quantization and constrain the entropy of the quantized representation to prevent spreading to all integers. Bounded scalar quantization (*i.e.*, FSQ), has been used to represent images with high fidelity (Mentzer et al. (2018) use $d{=}16, L{=}5$), and for "extreme compression" (Agustsson et al. (2019) used $d{=}5, L{=}5$). To the best of our knowledge, FSQ has not been used outside of compression. We note that the neural image compression field generally targets "high bitrate" reconstructions, and the challenge is to reduce the entropy of the complex representations, whereas in representation learning with VQ-VAE, the goal is usually the opposite: increase the entropy of a heavily constrained representation to maximally use it.

## 3 METHOD

We start with some high-level intuition. VQ defines a learnable Voronoi partition in the high-dimensional latent space of VQ-VAE, which leads to a complex non-linear partitioning of the VQ-VAE *input space* (*e.g.*, images). FSQ, by contrast, relies on a simple, fixed grid partition in a much lower-dimensional space. Intuitively this is feasible because VAEs have a relatively high model capacity in typical applications (see Sec. 2), and thus the non-linearity of VQ can be "absorbed" into

encoder and decoder, so that FSQ enables partitions of the VAE *input space* of similar complexity as VQ. We note that like in VQ-VAE based approaches, we train FSQ models in two stages: We first train a FSQ-VAE, and then a transformer on frozen FSQ-VAE representations.

## 3.1 FINITE SCALAR QUANTIZATION

Given a $d$-dimensional representation $z \in \mathbb{R}^d$, our goal is to quantize $z$ to a finite set of codewords. To this end, we first apply a bounding function $f$, and then round to integers. We chose $f$ such that each channel/entry in $\hat{z} = \text{round}(f(z))$ takes one of $L$ unique values (*e.g.*, $f : z \mapsto \lfloor L/2 \rfloor \tanh(z)$). Thereby, we have $\hat{z} \in \mathcal{C}$, where $\mathcal{C}$ is the *implied codebook*, given by the product of these per-channel codebook sets, with $|\mathcal{C}| = L^d$. The vectors in $\mathcal{C}$ can simply be enumerated leading to a bijection from any $\hat{z}$ to an integer in $\{1, \ldots, L^d\}$. Therefore, VQ can be replaced with FSQ in any neural network-related setup where VQ is commonly used, *e.g.*, to train transformers, after appropriately adapting the output and input dimension of the layers before and after VQ, respectively. We generalize the above exposition to the case where the $i$-th channel is mapped to $L_i$ values and get $|\mathcal{C}| = \prod_{i=1}^{d} L_i$.

We visualize FSQ in Fig. 1 (left) and in Fig. 2. Since quantization is performed by round to *integers*, supporting even $L$ requires an asymmetric $f$. We show the general $f$ used throughout this paper as code in App. A.1. To propagate gradients throughout the round operation, we use the STE throughout, replacing the gradients with 1. In ML frameworks, this can easily be implemented via the "stop gradient" (sg) operation as $\text{round\_ste} : x \mapsto x + \text{sg}(\text{round}(x) - x)$.

## 3.2 HYPERPARAMETERS

FSQ has the following hyper-parameters: the number of channels $d$ and the number of levels per channel, $\mathcal{L} = [L_1, \ldots, L_d]$. In most of our experiments, to obtain fair comparisons, we will choose target codebook sizes $|\mathcal{C}|$ based on the VQ codebooks we aim to replace with FSQ. However, various configurations of $d$ and $L_i$ can approximate a given $|\mathcal{C}|$ (*i.e.*, any $\mathcal{L}$ where $\prod_i L_i \approx |\mathcal{C}|$ is a candidate). We explore various configurations in our study, and find that not all choices lead to optimal results. However, we found a simple heuristic that performs well in all considered tasks: Use $L_i \geq 5 \, \forall i$. In Table 1 we tabulate $\mathcal{L}$ for common target $|\mathcal{C}|$.

## 3.3 PARAMETER COUNT

We note that FSQ has fewer parameters than VQ, since in VQ, a codebook of size $|\mathcal{C}| \cdot d$ is learned. For example, for a typical $|\mathcal{C}|=2^{12}=4096$ and $d=512$, this results in 2M parameters, which FSQ lacks. Additionally, since for FSQ, $d$ tends to be much smaller than for VQ (*e.g.*, $d=5$ for FSQ for this $|\mathcal{C}|$, see Tab. 1), the final encoder layer also has fewer parameters when training FSQ. To compensate for this, we explored adding more dense layers at the end of the VAE encoder, resp. at the start of the decoder, but found no further gains from doing so. *Thus, in all models in this paper, FSQ with the same codebook size has fewer parameters.*

## 4 EXPERIMENTS

## 4.1 REVIEW OF MASKGIT AND UVIM

We start with a brief review of MaskGIT (Chang et al., 2022) and UViM (**?**). In MaskGIT, the authors first train a (convolutional) VQ-GAN autoencoder (Esser et al., 2020) for reconstruction (Stage I). They then freeze the autoencoder, and train a masked transformer BERT-style (Devlin et al., 2018) to predict the quantized representations (Stage II): Given a representation $\hat{z}$, a fraction of tokens is randomly "masked out", *i.e.*, replaced with a special MASK token. The resulting sequence $\hat{z}_M$ is fed to a transformer in addition to a class token, and the transformer predicts a distribution for

| Target Size $|\mathcal{C}|$ | $2^8$ | $2^{10}$ | $2^{12}$ | $2^{14}$ | $2^{16}$ |
|---|---|---|---|---|---|
| Proposed $\mathcal{L}$ | $[8, 6, 5]$ | $[8, 5, 5, 5]$ | $[7, 5, 5, 5, 5]$ | $[8, 8, 8, 6, 5]$ | $[8, 8, 8, 5, 5, 5]$ |

Table 1: Recommended sets of FSQ levels $\mathcal{L}$ to approximately match a given codebook size $|\mathcal{C}|$.

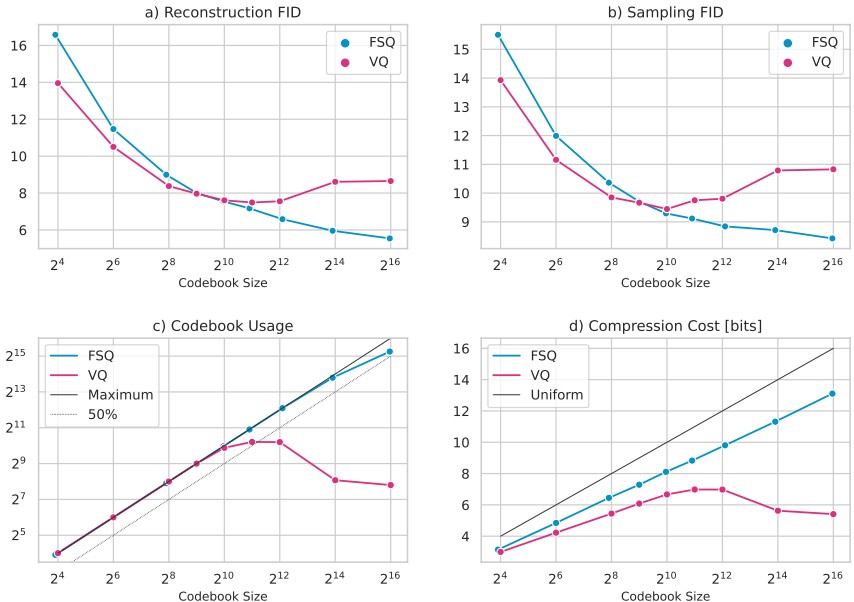

Figure 3: Characteristics and trade-offs for VQ and FSQ for $128 \times 128$ ImageNet. We see that Reconstruction FID correlates with codebook size for FSQ, and improves as we scale the codebook size. FSQ gets better Sampling FID and higher codebook usage for codebook size exceeding $2^{10}$, while the metrics start deteriorating for VQ.

each masked token. During inference, initially only MASK tokens along with the class token are fed to the transformer. Then, some of the token locations are selected based on prediction confidence, and corresponding tokens are sampled (see (Chang et al., 2022, Sec 3.2)). These tokens are used to replace mask tokens at the input, and the model is ran again, until all input tokens have been uncovered.

UViM (Kolesnikov et al., 2022) is a general architecture to tackle various (dense) prediction tasks in computer vision. In the first stage a transformer-based VQ-VAE is trained to model the label space of the target task. Optionally, both the VQ-VAE encoder and decoder can rely on the task input (RGB image for depth estimation and segmentation, grayscale image for colorization) as side information or "context", which was found beneficial for some tasks. In the second stage, an encoder-decoder transformer is trained to predict the dense label as quantized tokens produced by the VQ-VAE encoder, given the task input. For inference, a code is sampled autoregressively using the transformer conditioned on the input and then fed to the VQ-VAE decoder. The architecture is shared for the three tasks, but different weights are learned for each task.

## 4.2 CHARACTERISTICS AND TRADE-OFFS FOR VQ AND FSQ REPRESENTATIONS

We start with a study, where we train MaskGIT models on lower resolution $128 \times 128$ ImageNet images and for shorter time compared to the paper Chang et al. (2022) (100 epochs for Stage I, 200 epochs for Stage II. Please see Appendix A.4.1 for more hyperparameters). This allows us to sweep the codebook size and other hyperparameters. For VQ, we use the auxiliary entropy loss from MaskGIT, that aims to increase the entropy of the codebook (to increase utilization). We only sweep the codebook size. For FSQ, we explore various $d$ and $L_i$ to match these codebook sizes.

We track the following metrics: **Reconstruction FID**, the FID obtained by the GAN-trained autoencoder when the $50k$ validation images are fed through the quantized autoencoder. This is the FID that the Stage II transformer would achieve if it would perfectly model the data. We use the well established *ADM TensorFlow Suite* (Dhariwal & Nichol, 2023), which computes FID from 50k reconstructions w.r.t. the training set. **Codebook Usage**: The fraction of the codewords that are used at least once when encoding the validation set.

| Model | Source | CFG | Sampling FID$^\dagger\downarrow$ | Precision$^\dagger\uparrow$ | Recall$^\dagger\uparrow$ | Usage$\uparrow$ |
|---|---|---|---|---|---|---|
| MaskGIT (VQ) | Ours | 0.1 | 4.509 | 0.860 | 0.465 | 81% |
| MaskGIT (FSQ) | Ours | 0.2 | 4.534 | 0.864 | 0.453 | 100% |
| MaskGIT (VQ) | GitHub | - | 4.916 | 0.836 | 0.489 | |
| ADM (Dhariwal & Nichol, 2021) | | 1.5 | 4.59 | 0.83 | 0.52 | |

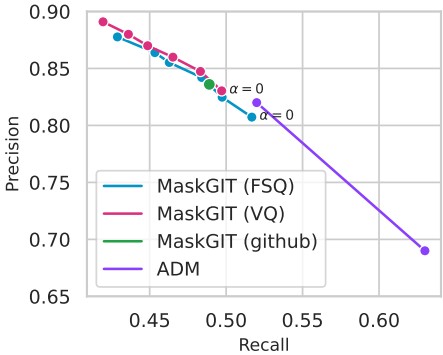 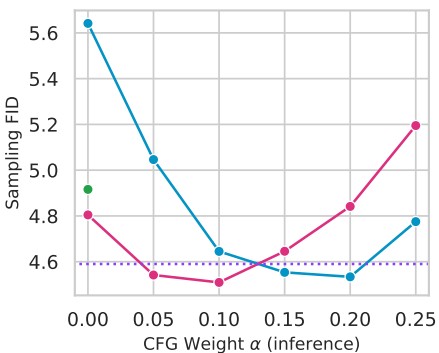

Figure 4: MASKGIT results on ImageNet 256. *Top:* We show the best classifier-free guidance (CFG) setting for each MaskGIT model. As a reference, we show the well established diffusion based ADM model (Dhariwal & Nichol, 2021). *Bottom Left:* Precision vs. Recall for various CFG weights. *Bottom Right:* Sampling FID for various CFG weights. We show ADM as a horizontal line, because the CFG weight 1.5 used for ADM is not comparable with our $\alpha$ in absolute terms. $^\dagger$We use the *ADM TensorFlow Suite* to evaluate all shown models, see text.

With the transformer trained in Stage II, we additionally report **Sampling FID**, the FID obtained when decoding representations $\hat{z}$ sampled (class-conditionally) with the transformer. We additionally propose studying **Compression Cost** as a proxy for how hard it is to model the discrete distribution underlying the representations (*i.e.*, modelling complexity): Note that any transformer that predicts a distribution over discrete codes can be used to *losslessly compress* the corresponding representation. For masked transformers, the only requirement is a deterministic masking schedule, that gradually uncovers the input. Using such a schedule, we can compress any $\hat{z}$ to bits, by pairing the transformer outputs with entropy coding. We use the deterministic masking schedule employed in M2T (Mentzer et al., 2023) and refer to Section 1 in that work for further details on the theory.

## 4.3 MASKGIT

We train MaskGIT models on ImageNet 256 based on the public GitHub code, training Stage I for 1M steps with batch size 512, and Stage II for 2.5M steps with batch size 256. For inference, we use 12 steps with the cosine to sample an image. Initial experiments with the public code showed a slight instability in the Stage II transformer loss, which we were able to mitigate by lower bounding the minimal masking ratio used during training. Please see Appendix A.4.3 for further details and hyper parameters. We train VQ with codebook size 1024 (10 bits) and the entropy loss, as in the published model. For FSQ, we use $\mathcal{L} = [8, 5, 5, 5]$ as suggested in Tab. 1.

Following the paper, we report **Sampling FID** as well as **Precision** and **Recall** (Sajjadi et al., 2018) to assess the quality of the generative model. Additionally, we also report **Codebook usage**. We again use the well-established *ADM TensorFlow Suite*, leading to an (ADM-)-FID-train of 4.916 for the official checkpoint published in the MaskGIT GitHub, vs. 6.19 reported in the MaskGIT paper.

Early experiments showed that FSQ lands at a different Precision & Recall point compared to VQ (FSQ had higher recall, lower precision). Inspired by the diffusion literature, we thus add classifier free guidance (**CFG**) (Ho & Salimans, 2022) to MaskGIT: During training, we replace 10% of the class labels with the MASK token to let the model learn the unconditional distribution. During inference, we interpolate logits: Let $l_c$ be the logits obtained when conditioning on the class label $c$, and $l_\emptyset$ be unconditional logits. During inference, we compute new logits $l' = l_c + \alpha(l_c - l_\emptyset)$,

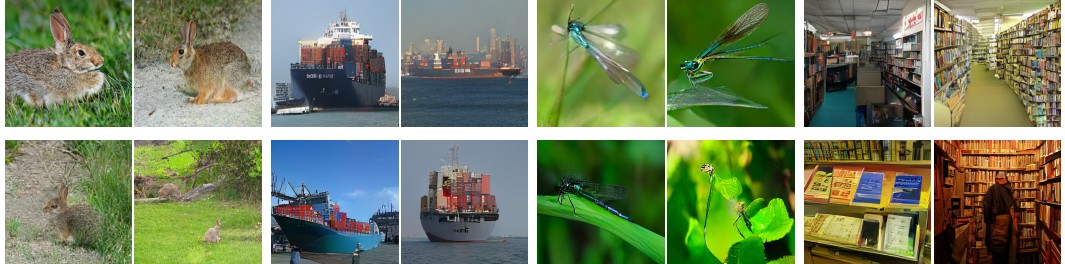

Figure 5: Non-cherry-picked samples from our FSQ (top) and VQ (bottom) MaskGIT models for 4 imagenet classes (`330, 320, 510, 454`). We show two samples per model per category. Both models get very comparable sample quality, as reflected by the metrics in Fig. 4.

where $\alpha$ is the CFG inference weight. Intuitively, this pulls the predicted distribution towards the unconditional one. We emphasize that this has previously been explored in the context of masked transformers, *e.g.*, by (Chang et al., 2023, Sec. 2.7).

## 4.4 UViM

We retrain the public UViM GitHub code for all three tasks (panoptic segmentation, depth estimation, colorization). As in the paper, we train each Stage II transformer 3 times, and report averaged metrics. For VQ, we use 4096 codewords (12 bits), and we use the codebook splitting (described below), as in the published results. We obtain similar metrics to what is reported in the GitHub repo, see Sec. 5. For FSQ, we use $\mathcal{L} = [7, 5, 5, 5, 5]$ from Tab. 1.

Following the UViM paper, we report panoptic quality (**PQ**) for panoptic segmentation, **RMSE** for depth estimation, and **FID-5k** for colorization. For all tasks, we use the evaluation suite provided by the UViM github repository. We refer to (Kolesnikov et al., 2022) for more details on these tasks and corresponding data sets.

We ablate the effect of VAE context input (*i.e.*, the RGB image, see above) on the performance of VQ and FSQ in the panoptic segmentation task. Further, we investigate the **codebook splitting** employed by UViM to avoid unused codewords in VQ-VAE. Specifically, they adopt the algorithm from Linde et al. (1980), where throughout training, unused vectors are detected. These are then replaced by splitting most frequently used embeddings into two new embeddings, adding noise to each. Since we observe training instabilities when deactivating codebook splitting in the panoptic segmentation task, we use the depth estimation task for this ablation.

## 5 RESULTS

### 5.1 TRADEOFF STUDY

In Fig. 3 we show the results for the trade-off study. On the x-axis, we always show the codebook size $|\mathcal{C}|$, representing the maximal amount of information the codebook can store. We observe the following:

**Codebook size correlates with Reconstruction FID for FSQ**   In Fig. 3 a), we see that as we increase the codebook size, the reconstruction FID for FSQ keeps improving. This is what one would expect from a compression perspective: as we have more bits to store information, we should get better reconstruction metrics. However, we see that VQ struggles with utilizing large codebooks (despite entropy regularization of the codes), and reconstruction FID achieves a minimum at $2^{11}$ codes, co-inciding with the point where the codebook usage starts decreasing (cf. Fig. 3 c)). We note that for low codebook sizes (Fig. 3 a), left), VQ marginally outperforms FSQ, likely owning to the its more expressive nature (see Contribution 3 in the Section 1).

**FSQ gets better Sampling FID**   A similar picture emerges in Fig. 3 b), where we see that the better Stage I behavior of FSQ translates to better Sampling FID as we scale the codebook.

| NYU Depth v2 | Source | RMSE$^\dagger$ ↓ | Codebook Usage |
|---|---|---|---|
| UViM (VQ) | Ours | $0.468 \pm 0.012$ | 99% |
| UViM (FSQ) | Ours | $0.473 \pm 0.012$ | 99% |
| UViM (VQ without splitting) | Ours | $0.490 \pm 0.0037$ | 0.78% |
| UViM (VQ) | GitHub | 0.463 | |
| DenseDepth (Alhashim & Wonka, 2018) | | 0.465 | |
| **COCO Panoptic** | Source | PQ$^\dagger$ ↑ | Codebook Usage |
| UViM (VQ) | Ours | $43.4 \pm 0.0008$ | 100% |
| UViM (FSQ) | Ours | $43.2 \pm 0.0014$ | 100% |
| UViM (VQ without context) | Ours | $39.0 \pm 0.0023$ | 99% |
| UViM (FSQ without context) | Ours | $40.2 \pm 0.0019$ | 99% |
| UViM (VQ) | GitHub | 43.1 | |
| DETR-R101 (Carion et al., 2020) | | 45.1 | |
| **ImageNet Colorization** | Source | FID-5k$^\dagger$ ↓ | Codebook Usage |
| UViM (VQ) | Ours | $16.90 \pm 0.056$ | 100% |
| UViM (FSQ) | Ours | $17.55 \pm 0.057$ | 100% |
| UViM (VQ) | Github | $16.99 \pm 0.057$ | |
| ColTran (Kumar et al., 2021) | | 19.37 | |

Table 2: UVIM results for the three tasks. For each, we show results in the corresponding metric averaged over three runs with std. dev. (as in UViM). We show the numbers reported by the reference GitHub repository, as well as one well established baseline per task. For our models, we show Codebook usage. For Depth Estimation, we train an ablation where we do not employ the codebook splitting in VQ. Overall, FSQ obtains competitive but marginally worse results on all tasks. $^\dagger$We use the UViM GitHub evaluation suite.

**FSQ gets high codebook usage**     In Fig. 3 c) we see that FSQ uses almost all codewords for a codebook size of $2^{14}=16k$, without employing any tricks. At the same time, VQ starts dropping below 50% usage for codebooks larger than $2^{11}$ and is not able to utilize more than $2^{10}$ codewords for larger codebooks. In contrast, for FSQ usage continues growing with more than $2^{15}$ codewords utilized for a codebook of size $2^{16}$.

**Diminishing gains from codebook scaling**     One might wonder whether just scaling the codebook size more would lead to ever lower sampling FID. However, as shown in Fig. 3 d), the compression cost of the representation keeps increasing. This indicates that the quantized representations get more complex to model for the transformer. Indeed, we see in Fig. 3 b) that the Sampling FID saturates for FSQ starting when using about $2^{12}$ codewords. We note that in general, for this task, the discrete distribution underlying the FSQ representations are slightly harder to model (as seen by the higher Compression Cost when training the same transformer on different VAEs, Fig. 3 d)). We also note how the Compression Cost for VQ correlates with the codebook usage: when the usage drops, the code becomes easier to model again. Similarly, within a model group (*i.e.*, considering only FSQ or VQ models), the compression cost is anti-correlated with sampling FID.

**Selecting the number of levels per channel** $\mathcal{L}$     In Appendix A.4.1 we also show the effect of different $\mathcal{L}$ on the Sampling FID. We find that $L_i < 5$ leads to subpar performance.

## 5.2 MASKGIT

In Fig. 4 we show the metrics for MaskGIT on $256 \times 256$ ImageNet. We sweep the CFG weight for both VQ and FSQ. The following can be observed:

**FSQ and VQ achieve comparable metrics and visual results**     Fig. 4 shows that both quantizers achieve very comparable FID, as well as precision and recall. To put the numbers in context, we show the well established diffusion-based ADM model (Dhariwal & Nichol, 2021). When inspecting the visual results in Fig. 5, we see that both quantizers lead to qualitatively similar samples. Motivated by the tradeoff study (sec. 5.1), we explored a larger codebook for these models, but did not observe further gains.

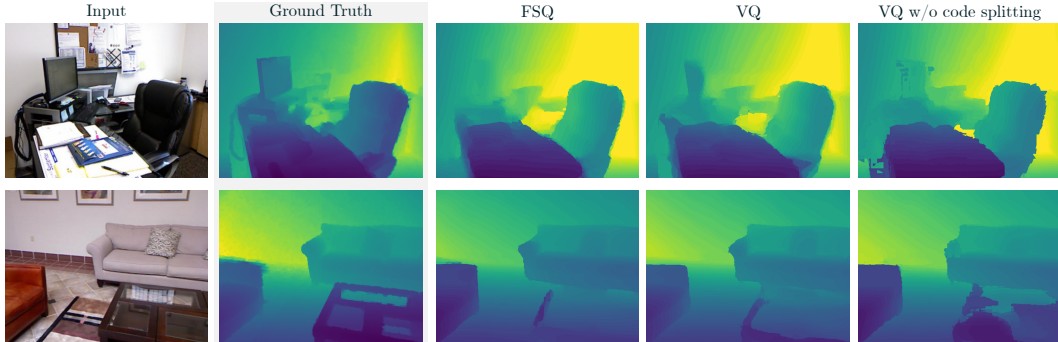

Figure 6: Samples from UViM for the depth estimation task. Other tasks in Appendix A.2. We observe that VQ and FSQ lead to comparable samples. VQ without splitting leads to jagged edges.

**Semantics**   It is commonly argued in the literature that the codebook in VQ-VAEs and VQ-GANs learns semantically meaningful codes. Yet, we see that we get similar samples from both VQ and FSQ, even though FSQ does not learn an explicit codebook (and thus has less parameters). We performed a small study to see whether either representation is more semantically meaningful than the other, shown in Appendix A.3. We found no evidence that a particular code represents a fixed visual concept in either quantizer. Indeed, both behave very similarly in that study.

**Precision-Recall trade-offs**   Note that precision is a measure for the "quality" of the samples, while recall measures the proportion of the true distribution that is covered by the samples (Sajjadi et al., 2018). When we sweep the CFG weight $\alpha$ during inference, we obtain models that cover a very similar space in Precision & Recall (bottom, left), and that obtain very similar minimal FID (bottom, right).

## 5.3   UViM

Table 2 shows the results for the three tasks trained with UViM along with some baselines from the literature.

**FSQ is competitive with VQ on all tasks**   We can see that across all tasks, FSQ obtains competitive metrics compared to VQ. This is also reflected in the visual results shown in Fig. 6 (for depth estimation) and App. A.2 (for panoptic segemenation and colorization).

**FSQ performs better in absence of side information (context)**   Table 2 also shows removing the VAE context in UViM (panoptic segmentation), *i.e.*, removing the original RGB image input to the VAE encoder and decoder (see Sec. 4.1). In this setting, both the FSQ and VQ-based models obtain lower PQ numbers than with context, but the performance of the FSQ-based model degrades less.

**FSQ does not rely on codebook splitting**   We explore disabling the codebook splitting on the *NYU Depth* task, and we observe signficantly worse RMSE, while Codebook usage drops by more than two orders of magnitude to 0.78%. In the predictions, we observe jagged edges, see Fig. 6 (right most column). At the same time, FSQ does not rely on any auxiliary algorithms to obtain 99% codebook usage.

## 6   CONCLUSION

In this work, we showed that we can replace the vector quantizer in VQ-VAEs with a simple scalar quantization scheme, where the representation is projected to very few dimensions which are bounded and rounded. We studied and compared the behavior of FSQ and VQ as a function of the codebook size and observed that FSQ achieves much better codebook utilization for large codebook sizes. Despite the much more constrained setup, we were able to obtain comparable metrics on image generation with MaskGIT, and dense computer vision tasks with UViM. We hope future work will explore FSQ in even more applications.

**Reproducibility**   We refer to Section A.1 for reference code.

**Ethics Statement**   This work proposes a drop-in replacement for VQ, and can thus be applied in all domains where VQ is used. A domain where care w.r.t. biases has to be taken is generative models. However, no new ethical concern arises from our method that would not be a concern for VQ-based methods.

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
