# A    APPENDIX — FINITE SCALAR QUANTIZATION: VQ-VAE MADE SIMPLE

## A.1    CODE

We refer to the MaskGIT GitHub and the UViM GitHub for the model code used in this paper. The FSQ method is implemented in full generality for Jax (Bradbury et al., 2018) in the following listing:

```python
def round_ste(z):
  """Round with straight through gradients."""
  zhat = jnp.round(z)
  return z + jax.lax.stop_gradient(zhat - z)

class FSQ:
  def __init__(self, levels: list[int]):
    self._levels = levels
    self._levels_np = np.asarray(levels)
    self._basis = np.concatenate(
        ([1], np.cumprod(self._levels_np[:-1]))
    ).astype(np.uint32)

    codebook_size = np.prod(levels)
    self.implicit_codebook = self.indexes_to_codes(
        np.arange(codebook_size))

  def bound(self, z):
    """Bound `z`, an array of shape (..., d)."""
    eps = 1e-3
    half_l = (self._levels_np - 1) * (1 - eps) / 2
    offset = jnp.where(self._levels_np % 2 == 1, 0.0, 0.5)
    shift = jnp.tan(offset / half_l)
    return jnp.tanh(z + shift) * half_l - offset

  def quantize(self, z):
    """Quanitzes z, returns quantized zhat, same shape as z."""
    quantized = round_ste(self.bound(z))
    half_width = self._levels_np // 2  # Renormalize to [-1, 1].
    return quantized / half_width

  def _scale_and_shift(self, zhat_normalized):
    half_width = self._levels_np // 2
    return (zhat_normalized * half_width) + half_width

  def _scale_and_shift_inverse(self, zhat):
    half_width = self._levels_np // 2
    return (zhat - half_width) / half_width

  def codes_to_indexes(self, zhat):
    """Converts a `code` to an index in the codebook."""
    assert zhat.shape[-1] == len(self._levels)
    zhat = self._scale_and_shift(zhat)
    return (zhat * self._basis).sum(axis=-1).astype(jnp.uint32)

  def indexes_to_codes(self, indices):
    """Inverse of `indexes_to_codes`."""
    indices = indices[..., jnp.newaxis]
    codes_non_centered = np.mod(
        np.floor_divide(indices, self._basis), self._levels_np
    )
    return self._scale_and_shift_inverse(codes_non_centered
```

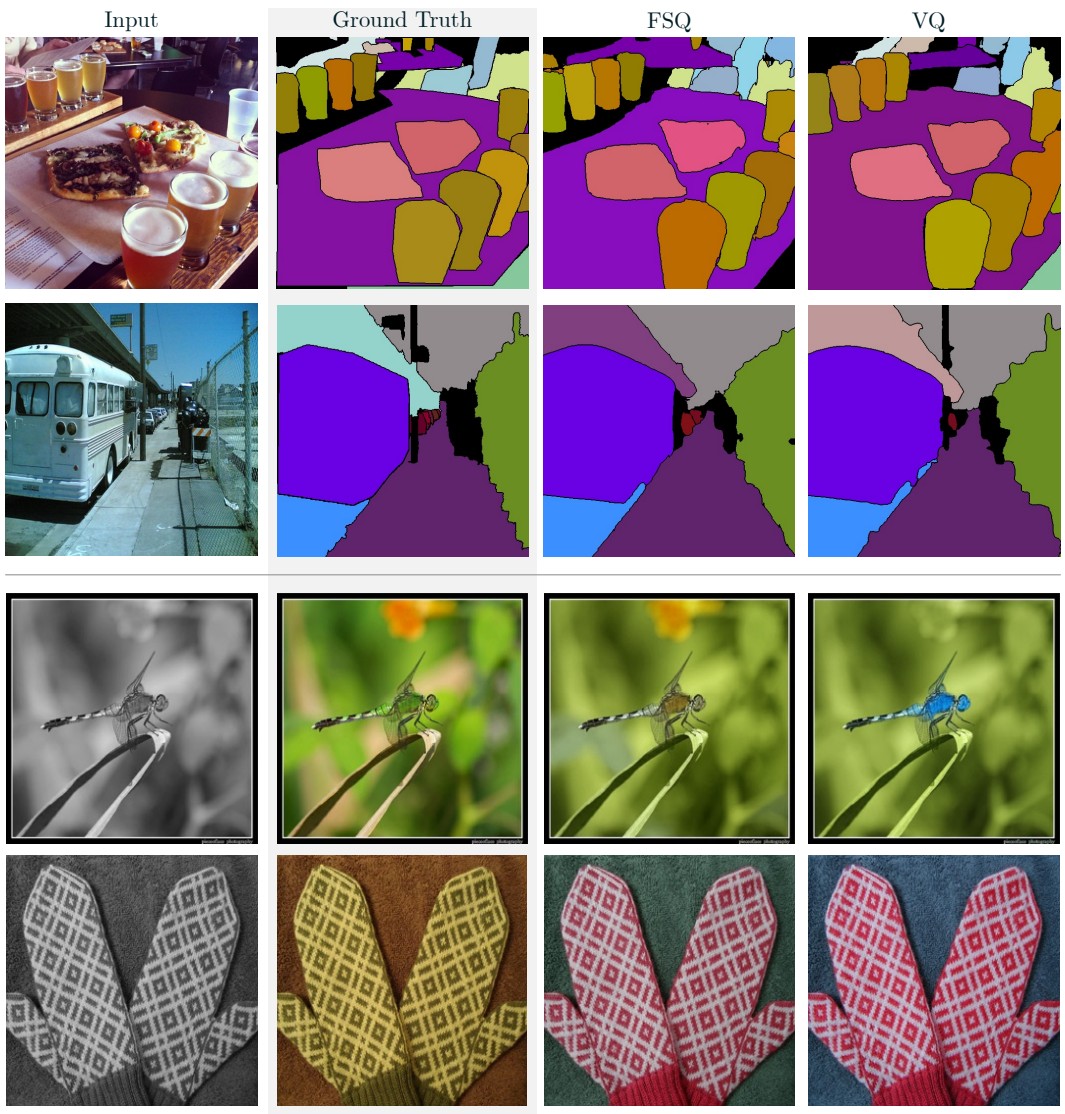

Figure 7: Visualization for panoptic segmentation (first two rows) and colorization (last two rows).

## A.2 ADDITIONAL UVIM VISUALS

We show visual results for segmentation and colorization in Fig. 7. Results for depth estimation are in Fig. 6 in the main text.

|  Top  |  Bottom  |  Stited Pixels  |  Stitched FSQ  |  Stitched VQ  |

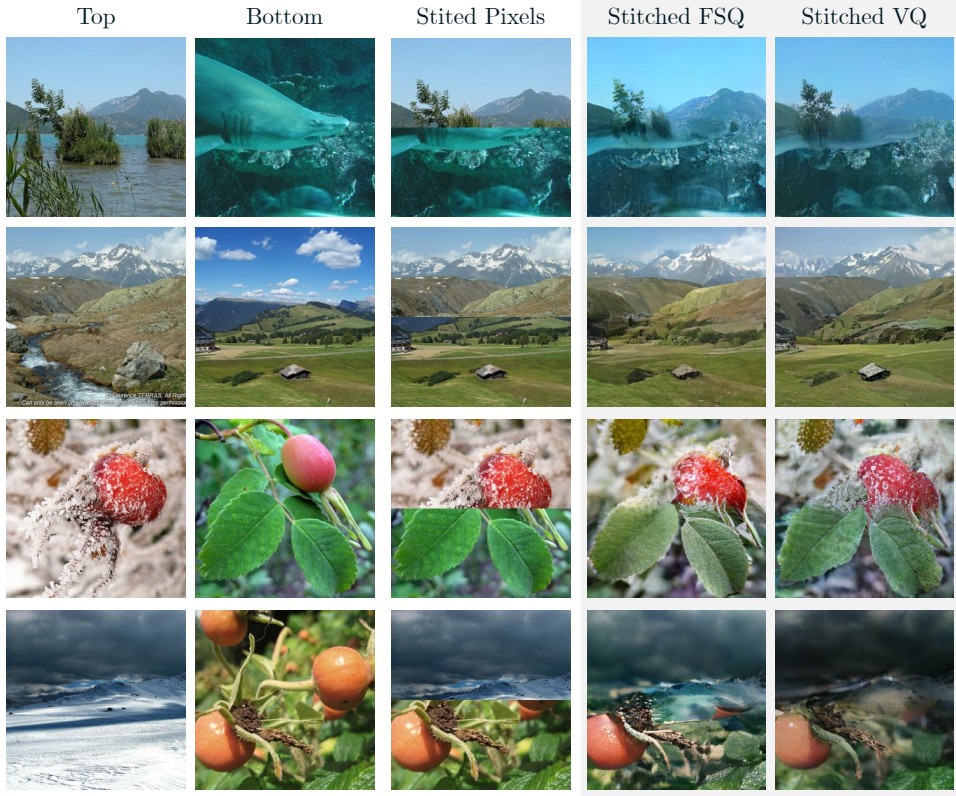

Figure 8: Analyzing representations: we take two random images A, B from the validation set (first two columns). We compare stitching the top half of A to the bottom half of B in pixel space (center) to stitching the corresponding representations obtained by the FSQ-GAN and VQ-GAN (last two columns) in latent space. Note how the GAN decoder maps the sharp transitions in representation space to smooth transitions in pixel-space.

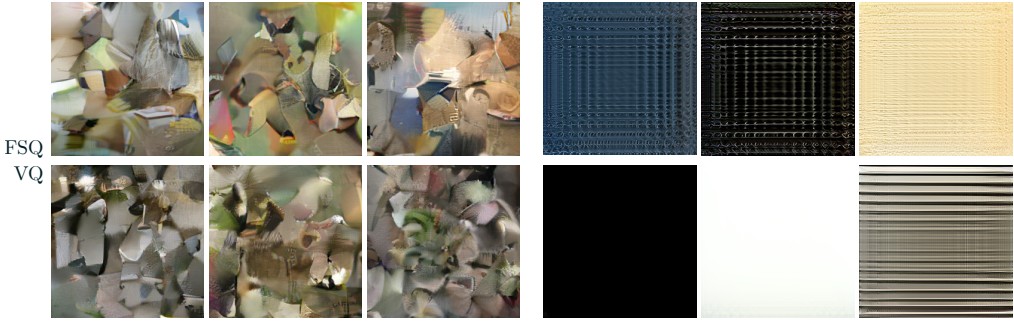

Figure 9: Analysing "fake" representations: *Left 3 columns*: randomly sampling codes according to the marginal histogram, for FSQ (top) and VQ (bottom). *Right 3 columns*: Creating a representation sharing code across all spatial location, where we pick the 3 most common codes according to the marginal histogram (left-to-right).

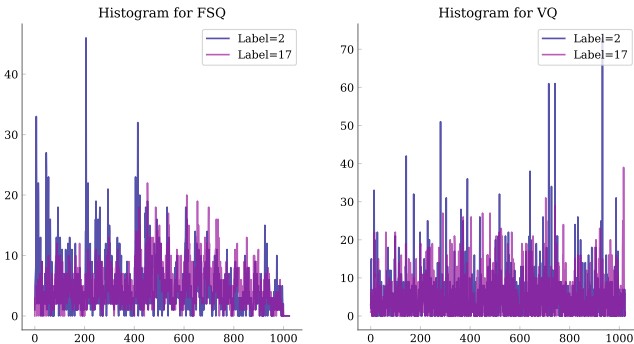

Figure 10: Comparing the codewords used by encoding 20 images of two labels (2 "Shark", 17 "Jay") and looking at the histograms. On the left, we show FSQ, on the right VQ. We see that both autoencders use most codwords to represent both images.

## A.3 VISUALIZING VQ AND FSQ

We are interested in what the representations of our MaskGIT autoencoders store. In Fig. 9, we visualize "average" representations: for each autoencoder (FSQ-GAN and VQ-GAN), we create marginal histograms by encoding the entire ImageNet validation set. We then sample 3 $16{\times}16$ representations from each histogram, and decode the representation with the resp. decoders. Both produce similar "soup of patches". We also visualize representations sharing a single code across all spatial locations.

We further stitch together representations obtained by encoding real images in Fig. 8. We see that both decoders smoothly blend the the stitched representations when decoding to RGB space.

In Fig. 10, we show how encoding two different classes of images leads to a big overlap in indices, and in Fig. 11 we show how editing the FSQ representation affects textures in the output.

Overall, this investigation seems to imply that individual codes do not learn very abstract concepts. Instead it is the combination of codes decoder weights which determine the final RGB image.

Finally, we show reconstructions of feeding 5 validation images through our autoencoders in Fig. 12. Here, only the autoencoder is used.

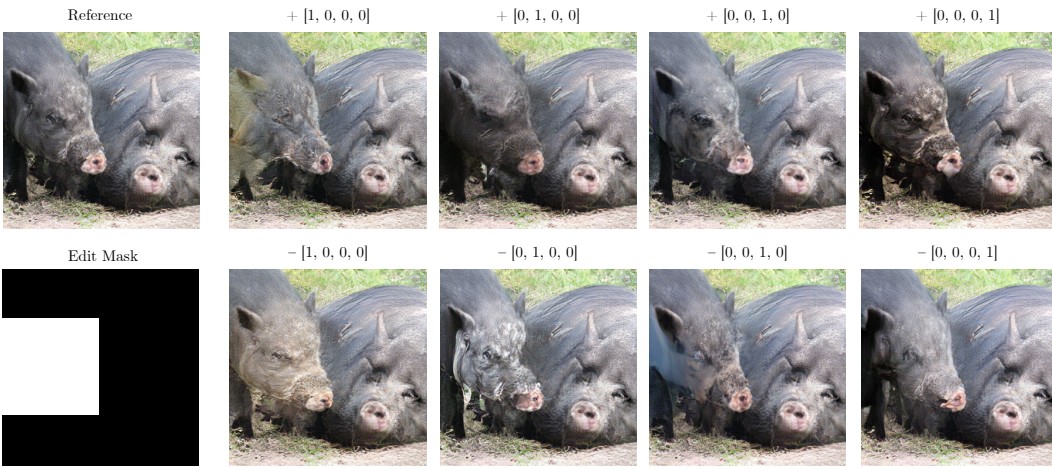

Figure 11: We show how adding a vector to the FSQ representation changes the reconstruction. For this image, we encode the reference image into a representation $\hat{z}$ using the FSQ encoder. This representation ahs $d = 4$ channels. Then we add, *e.g.*, [1, 0, 0, 0] to each vector, but only for the locations given by the edit mask. We see that this changes the texture, but not the semantic content.

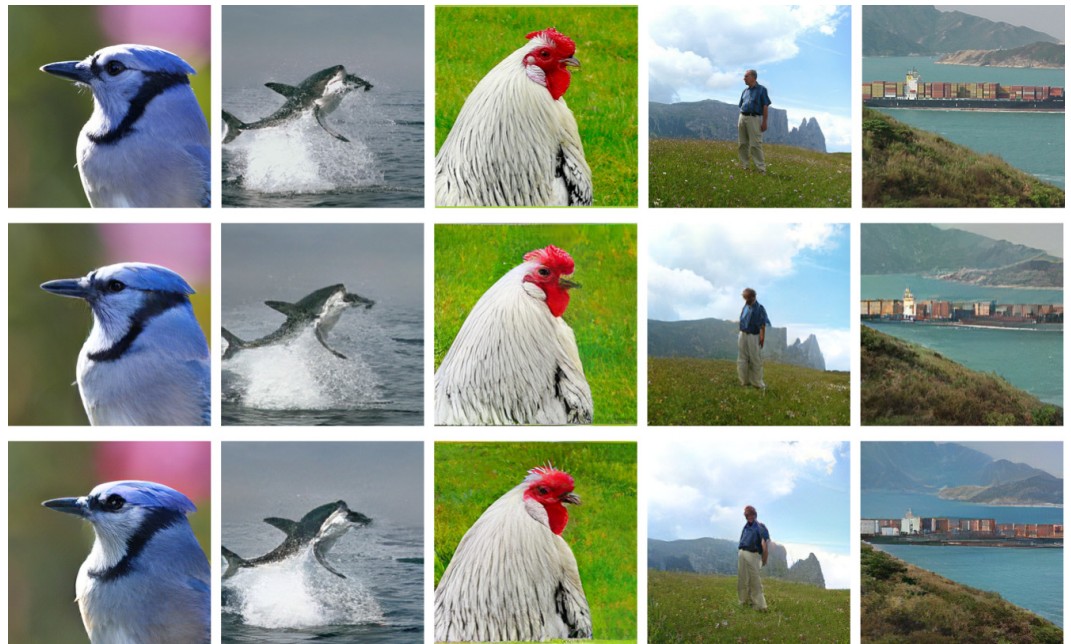

Figure 12: Reconstructions from the validation set. *First row*: Ground truth validation image. *Second row*: FSQ reconstruction. *Third row*: VQ reconstruction.

## A.4 Training Details

### A.4.1 Tradeoff Study

We use MaskGIT and train stages I and II on $128 \times 128$ ImageNet. We explore a range of configurations for the quantization levels $\mathcal{L}$ in FSQ models and show the results in Fig. 13. We find that $L_i \geq 5$ leads to the best performance. Intuitively, this is because of the straight through estimation (STE) that we use: for this estimation, we approximate the gradient of the round operation with the gradient of the identity function (*i.e.*, the line y=x). This approximation is very bad for L=2, when the quantizer becomes a step function, but gets better and better as we increase L. Motivated by this we recommend the following codebook sizes for $\mathcal{L}$ for FSQ:

| $2^4$ | $2^6$ | $2^8$ | $2^9$ | $2^{10}$ | $2^{11}$ | $2^{12}$ | $2^{14}$ | $2^{16}$ |
|-------|-------|-------|-------|----------|----------|----------|----------|----------|
| $[5,3]$ | $[8,8]$ | $[8,6,5]$ | $[8,8,8]$ | $[8,5,5,5]$ | $[8,8,6,5]$ | $[7,5,5,5]$ | $[8,8,8,6,5]$ | $[8,8,8,5,5,5]$ |

We use 100 epochs for Stage I, split into $\approx 500k$ steps of batch size 256, and 200 epochs split into $\approx 1M$ steps for Stage II, also using batch size 256.

As mentioned in the main text, we employ a minimal masking ratio to stabilize Stage II training described in Sec A.4.2. All other hyperparameters are copied from the `vqgan_config.py` and `maskgit_class_cond_config.py` configs from the MaskGIT GitHub. We emphasize that for VQ we use the entropy loss from MaskGIT with weight 0.1.

### A.4.2 Lowerbounding the MaskGIT masking ratio

MaskGIT uses a cosine schedule to sample masking ratios during training, where first a ratio $r \sim U[0,1]$ is sampled, and then $N_M = \lceil \cos(\pi/2(1-r))S \rceil$ randomly selected tokens are masked for each example in the mini batch. $S$ is the sequence length, which is $16^2 = 256$ for models trained on ImageNet 256. We found that this causes instability, likely because there are training steps, where $N_M = 1$, *i.e.*, only one token is masked, and we only get a loss from the corresponding prediction. Instead, we lower-bound $r$ to $r_{\min} = 1 - (\arccos(0.45)2/\pi)$, which results in $N_M > 0.45S$ for every training step. We later explored various alternatives to 0.45 and found that any value above 0.2 helps with stabilization, but use 0.45 throughout.

### A.4.3 MaskGIT on ImageNet256

Again, we base all experiments on the `vqgan_config.py` and `maskgit_class_cond_config.py` configs from the MaskGIT GitHub repo. To speed up iteration, we change the VQGAN config to use 1M steps with batch size 512 (for Stage I), instead of 2M steps with batch size 256. We again lower bound the masking ratio as described in Sec. A.4.2. We do not change anything else from the above mentioned configs.

### A.4.4 UViM

We follow the UViM GitHub with very minimal changes: for the VAE, we train a separate FSQ-VAE using the `vqvae_*.py` configs from the repo adapted to use $\mathcal{L} = [7,5,5,5,5]$. We do not change the transformer configs (`train_*.py` configs in the repo).

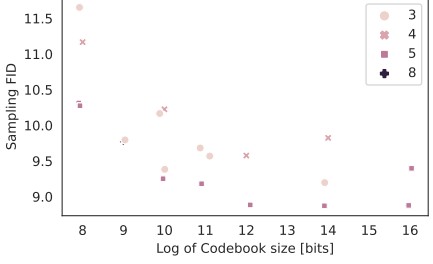

Figure 13: Exploring different configurations of quantization levels per channel $\mathcal{L}$. The color and marker indicate the smallest $L_i$ used for a given model (see legend).

We benchmarked the inference of sampling 8 images from our FSQ and VQ models. For this, we run the MaskGIT model for 16 steps, thereby obtaining indices, convert the indices into vectors and run the VAE decoders. We use a single core of a TPUv3, and report the mean over 10 trials. In each trial, we sample 8 images in parallel. We report the results in Tab. 3. Both quantizers lead to nearly identical runtimes.

|  | FSQ | VQ |
| --- | --- | --- |
| Sample 8 images | 678ms | 678ms |

Table 3:  Benchmarking our ImageNet models on a TPUv3. Added for v2