# OpenReview forum: "Finite Scalar Quantization: VQ-VAE Made Simple"
_ICLR.cc/2024/Conference — ICLR 2024 poster_

### Official Review · Reviewer_zMky · 2023-10-18

**Soundness:** 3 good
**Presentation:** 3 good
**Contribution:** 2 fair
**Rating:** 6
**Confidence:** 4

**Summary:**

This paper proposes a simple method called finite scalar quantization (FSQ) to replace vector quantization (VQ) in VQ-VAEs. FSQ projects the representation to very few dimensions (typically less than 10), with each dimension quantized to a fixed set of values. This implies a codebook of the same size as in VQ. Results show that as codebook size increases, FSQ achieves better reconstruction metrics and higher codebook utilization, while these metrics deteriorate for VQ.

**Strengths:**

1. This paper is written clearly and proposes a straightforward solution FSQ.
2. Using scalar quantization to replace vector quantization in VQ, FSQ method avoids the codebook collapse problem in VQ and does not need auxiliary losses.
3. Sufficient and concrete experiments demonstrate the effectiveness of the proposed method.
4. The authors make detailed comparisons between the performance of FSQ and VQ.

**Weaknesses:**

Although I believe this paper meets the bar of acceptance, I still have a few concerns:

1. FSQ's expressive capability is slightly weaker than VQ, and its performance is slightly worse with small codebooks.

2. The paper lacks analysis on the speedup and compression rate brought by the quantization of FSQ.

3. FSQ also faces difficulties in finding proper quantization hyperparameters (dimensions and number of quantization levels).

**Questions:**

I hope the authors can add some intuitive numerical comparisons to demonstrate the inference speedup brought by FSQ, and emphasize that FSQ does not need the parameters for the codebook.

Additionally, from the results in the paper, we can see that the metrics of FSQ only show significant advantages when the codebook size is greater than 2^10. However, wouldn't such a large size have a low cost-effectiveness for the second stage (I mean the trade-off between parameter size and FID, where a larger codebook may require a larger model to handle)? This will affect the model's practicality.

---

> ### Author Response · Authors · 2023-11-14
>
> Thanks for the review, and the kind words in the Strenghts section!
>
> Regarding the capability difference, and small codebook differences: We agree, and we note this in the text (Sec 5.1, “Codebook size correlates with Reconstruction FID for FSQ”). However, we also note that FSQ has more favorable scaling behavior (Fig 3), which we see as a very promising direction for future work: we note that 1024 (2^10) is not very large, compared to language modeling, where people routinely use 32k+ codewords. In general, we think that approaches relying on VQ mostly did not further scale the codebook because VQ becomes unstable as one increases the codebook, but a priori, if one can afford a big enough model in the second stage, it should be desirable to increase the codebook. We see this empirically for FSQ in Fig. 3, where scaling the codebook improves sampling FID, without requiring a larger stage 2 model.
>
> Regarding speed (weakness 2 and question 1): we note that we do not claim inference speedups from FSQ in the paper. We note that FSQ is only faster when calculating indices (since VQ requires to compare a vector with every possible vector in the codebook, it has O(|C|) cost of computing an index, whereas FSQ has O(d) cost, as we just need to run a single dot product (see Sec. A.1)). For inference, when we sample from the transformer and then feed the result to the VAE decoder, both methods have the same theoretical runtime. We validated this with a small benchmark added to Section A.5 in the updated PDF.
>
> Regarding the compression rate, we see that FSQ can utilize the codebook better than VQ (Fig 3, d), and thus uses more bits when used as a compressor. This results in better sampling FID, since the transformer can create high information representations (Fig 3, b).
>
> Regarding the number of quantization levels, we found the rule to use L >= 5 to work well across tasks, and report good choices for d and L for different target codebook sizes in Table 1. We note that the principle of using L>=5 is validated empirically, see Sec A.4.1 Trade-off study, where we found that codebooks with L<5 also work well, but L>=5 performs a bit better (the difference is <1 FID). Intuitively, we think this is because of the straight through estimation (STE) that we use: for this estimation, we approximate the gradient of the round operation with the gradient of the identity function. This approximation is very bad for L=2, when the quantizer becomes a step function, but gets better and better as we increase L, with diminishing returns as L further increases. We added this intuition to A.4.1, see updated PDF (changes in pink)

---

> > ### Comment · Reviewer_zMky · 2023-11-21
> >
> > Thank you for your response. Your answers have resolved most of my concerns. I am willing to keep my score unchanged (6) and am very inclined to accept.

---

### Official Review · Reviewer_fsTh · 2023-10-24

**Soundness:** 3 good
**Presentation:** 3 good
**Contribution:** 2 fair
**Rating:** 6
**Confidence:** 4

**Summary:**

The paper proposes Finite Scalar Quantization (FSQ) to replace the popular VQ-VAE.
Instead of performing cumbersome VQ-VAE optimizations (multiple losses, handling codebook underutilization, etc.), FSQ performs a simple quantization (rounding) of the learned embedding, optimized via the standard STE.
The method is able to reach FSQ performance on several tasks.

**Strengths:**

1. The paper is clear
2. The approach is somehow original for generative models
3. The method is simple (appears to be much simpler than the VQ-VAE cumbersome training and optimization) with less model capacity and computational complexity
4. Yet the method reaches good performance.

**Weaknesses:**

1. The main weakness lies in the originality/novelty of the method which appears to be a compressive autoencoder (e.g., [1,2]) applied to generative tasks. As the Method section demonstrates, there are very few (no) technical/theoretical contributions.

2. The results are still (mostly slightly) worse than VQ.

3. The method obviously lacks compression performance since the quantized vectors' indices are insufficient/irrelevant in this case.

[1] *Lossy Image Compression with Compressive Autoencoders*

[2] *Weakly Supervised Discovery of Semantic Attributes* (and this is not a compression method)

**Questions:**

1. There is an obvious difference between the FSQ and VQ-VAE functionals (rounding vs L2 distance). It would be interesting to see the difference in the quality of the representations (e.g., same class samples should be close(r with FSQ?) in their representations) and not only regarding the optimized metrics.

2. Following 1., it is not clear how the codebook is seen as implicit and not simply defined by the hypercube's samples.

3. How do you explain that extending the model capacity (Sec 3.3) does not result in improvement, especially since the method is still worse than VQ-VAE?

---

> ### Author Response · Authors · 2023-11-14
>
> Thanks for the review, and the strengths you point out.
>
> Regarding novelty: first, we note the link to compressive autoencoders in the paper (Sec 1, 2) and do not claim novelty on the scalar quantizer itself. However, as we note in the general reply, our contribution lies in considering the implied product codebook of FSQ: “We’d like to emphasize that prior to this paper, no one was thinking of using finite scalar quantization (FSQ) as a drop-in for VQ in generative image models or for dense vision tasks. Our key insight is to consider the implicit product codebook created by scalar quantization if you bound each channel, which allows training large generative transformers like on VQ representations (i.e., by bounding each of the $d$ channels to $L$ levels, we get a product codebook of $L^d$ codewords). While it might seem obvious after reading our paper, a priori, it is very surprising that FSQ obtains such competitive results compared to VQ -- despite not using any of its auxiliary losses and tricks.”
>
> Regarding comparison to VQ: we note that in Fig. 3, we see that FSQ scales well to high-bit codebooks. Arguably, this is where the future research lies, as small codebooks are becoming less and less important. We think that people focused on eg 1024 codewords, because VQ did not scale beyond that. Our contribution lies in showing that our simple approach can be highly competitive even though VQ is highly optimized and much more challenging to train.
>
> Regarding “The method obviously lacks compression performance since the quantized vectors' indices are insufficient/irrelevant in this case.”: we note that in this work we do not target compression, which typically operates at much higher bitrates (note that our codebooks of e.g. 10 bits (1024 codewords) correspond to 0.03 bits per pixel without any entropy coding). One can increase the codebook size and eventually obtain a competitive compression method, similar to what was studied in the compression literature (see “Neural Compression” references in Section 2). To improve rate-distortion performance, one probably wants to leverage entropy coding to get an even more compressed bitstream. However, we note that we target generative downstream tasks, and we see in Fig 3d that FSQ can utilize the codebook better than VQ. This is desirable for the tasks we study, and results in better sampling FID (Fig 3b).
>
> Q1) Thanks for this suggestion, this also made us curious. We calculated the representations for 3 images: 2 from class A, and 1 from class B. When calculating the MSE between the two images in class A in FSQ representation space, we obtain 1.5e5. When calculating the MSE between one image from A and the image from B, we obtain 1.4e5. For VQ, the same story emerges, we get 1.67e5 for in-class, and 1.53e5 for between-class. In both cases, see that images from the same class do not lie closer in the representation space. This is in line with what we already report in Section A.3., where we concluded that the representations of both FSQ and VQ mostly store medium level textures. Inspired by your comment, we added a few results to Section A.3:
> - Fig 10, which shows the codewords used for two different imagenet classes when encoding 20 examples. As it turns out, the codewords seem to represent textures more than semantic concepts: both models use most codewords to represent very different classes.  This explains that we do not see a close relationship when calculating MSE above.
> - Fig 11, where we show what happens when adding offsets to the FSQ channels. Again, only low level texture details change.
>
> Overall, both VQ and FSQ seem to store mostly medium level texture detail. Note that both approaches store 1 codeword per 16x16 pixel patch.
>
> Q2) we call the codebook "implicit" because it is defined by the product of the per-channel codebook (eg for L=3, d=2, the per channel codebook is {-1, 0, 1}, and the implied codebook is given by {(-1, -1), (-1, 0), (-1, 1), …}). As you point out, this could be implemented by using VQ with a fixed codebook set to the hyper cube, but in our approach, we never have to construct this codebook: during encoding, the index into the implicit codebook can be calculated with a single dot product (see Sec. A.1), and we thus call it implicit.
>
> Q3) This is likely because in the experiment we ran, the capacity we add is comparatively small, e.g., VQ codebooks for MaskGIT store 1024 * 256 parameters, which is <0.1% of the 227M parameters of the entire model (VAE + transformer). In general, it should help to add a substantial amount of parameters in exchange for more memory and compute requirements.

---

### Official Review · Reviewer_HYXB · 2023-11-01

**Soundness:** 3 good
**Presentation:** 2 fair
**Contribution:** 3 good
**Rating:** 8
**Confidence:** 3

**Summary:**

This paper proposes a new type of discrete tokens used in VQ-VAE named finite scalar quantization (FSQ). The dimension of the VQ-VAE representation is decreased to a low number and each dimension can only take a few scalars. With this design, FSQ is able to achieve the same size of the codebook used in original VQ-VAE with far less parameters. In addition, FSQ does not require complex design like entropy penalty as well. By using these FSQ representation, the models using VQ-VAE can also be trained with FSQ, such as image generation, multimodal generation, and dense prediction computer vision tasks. Experiment results show that FSQ has competitive performance with original VQ-VAE and prove the expressive discrete representations learned by FSQ.

**Strengths:**

* S1: The method is effective and easy to implement, since no extra complex design is needed.
* S2: The writing of the FSQ design is easy to follow.
* S3: The experiments are extensive. The authors demonstrate the efficiency of the proposed method under various settings.

**Weaknesses:**

* W1: The relation between VQ-VAE and FSQ is confusing. Specifically, I do not understand how the FSQ model is trained. Is the FSQ model trained from scratch using the VAE architecture, or is the FSQ model transformed from a pretrained VQ-VAE model in some way. Since MaskGIT is a two stage method, where the first stage is to train a discrete tokenizer, I will regard FSQ as a model trained from scratch.
* W2: The principle of desigining the shape of the codebook is not given. It seems that the number of levels per channel of the recommended shape is uniform and larger than 5. But there is not a intuitive explanation on why.

**Questions:**

Since the paper does not contain the training process of FSQ, I am a little confused about the setting. Here are some questions:

* Q1: As I mentioned in the weakness part, is FSQ used in MaskGIT and UViM trained from scratch? If yes, what is the model architecture and the other hyperparameters like training epochs? And what is the reconstruction and generation ability of FSQ used as a VAE?

* Q2: There is another concurrent work named Lookup Free Quantization [1], which also aims to compress the codebook. Their method directly let the dimension of codebook features become zero, which can also support a larger vocabulary size. Can the authors compare the proposed method with theirs?

I will be very happy to raise my score if these concerns are addresed.

[1] Language Model Beats Diffusion -- Tokenizer is Key to Visual Generation, arxiv 2310.05737

---

> ### Author Response · Authors · 2023-11-14
>
> Thanks for the review, and for the kind words. Regarding the weaknesses and questions:
>
> W1) Apologies for the confusion around training, what is happening is that we train what you could call a “FSQ-VAE” instead of a VQ-VAE. Then we train transformers in a second stage. Your understanding thus is correct (“Since MaskGIT is a two stage method, where the first stage is to train a discrete tokenizer, I will regard FSQ as a model trained from scratch.”) -- We updated the PDF with the following: “We note that like in VQ-VAE based approaches, we train FSQ models in two stages: We first train a FSQ-VAE, and then a transformer on frozen FSQ-VAE representations.” (Sec 3, change highlighted in pink).
>
> W2) The principle of using L>=5 is validated empirically, see Sec A.4.1 Trade-off study, where we found that codebooks with L<5 also work well, but L>=5 performs a bit better (the difference is <1 FID). Intuitively, we think this is because of the straight through estimation (STE) that we use: for this estimation, we approximate the gradient of the round operation with the gradient of the identity function. This approximation is very bad for L=2, when the quantizer becomes a step function, but gets better and better as we increase L, with diminishing returns as L further increases. We added this intuition to A.4.1, see updated PDF (changes in pink).
>
> Q1) Indeed, our FSQ-VAEs are trained from scratch. We use virtually the same hyper parameters as the baseline VQ based methods, relying on the public code. We clarified this in the updated draft in A.4.3 and the newly added A.4.4. With the respective GitHub repos and our code in Sec A.1., our paper can be reproduced. Regarding the reconstruction ability: thanks for this suggestion, we added Fig 13 to the paper (in the supplementary material). Both VQ and FSQ behave very similarly in terms of reconstructions, we see FSQ is slightly more consistent with the input in terms of color.
>
> Q2) Thanks for pointing out this concurrent work. We note that the authors still rely on an additional entropy maximization loss (Equation 5 in their paper), whereas FSQ does not use any auxiliary losses. It will be interesting to see if future work sees a benefit from adding an entropy maximization loss to our method.

---

> ### Comment · Reviewer_HYXB · 2023-11-15
> **Thanks for the detailed explanation! I will raise my score.**
>
> Thanks for the detailed explanation. The illustration on the two-stage training address my concern. I love the potential of this work to scale up the size of the codebook. Thus I will raise my score from 5 to 8.
>
> Additionally, there is another work named MAGE [1] which also relies on discrete tokenization to reach supreme classification performance and SOTA generation ability among MIM methods. MAGE is a variant of MAE [2] where the input and prediction target of MAGE are both discrete tokens. I believe this work will shed some lights on the improvement of MAGE.
>
> [1] MAGE: MAsked Generative Encoder to Unify Representation Learning and Image Synthesis. CVPR 2023.
> [2] Masked Autoencoders Are Scalable Vision Learners. CVPR 2022.

---

### Official Review · Reviewer_Hp3C · 2023-11-06

**Soundness:** 3 good
**Presentation:** 3 good
**Contribution:** 3 good
**Rating:** 6
**Confidence:** 2

**Summary:**

This work considers replacing the quantization layer of vector quantized variational autoencoders (VQ-VAE) with a scalar quantization strategy. The goal of VQ-VAE is discrete representation learning, i.e., we want to embed a datapoint in the latent space using an encoder network, and we want the representation in the latent space to belong to a discrete set of finite values (referred to as the codebook). This codebook is learnt by jointly training the encoder and decoder network.

Traditional VQ-VAE approaches consists of a quantization layer, in which the codebook (or the quantization points) is done using K-means clustering or exponential moving average. These approaches have some issues such as codebook collapse, in which the learnt codebook is a very small set of points compared to the latent space. The strategy proposed in this paper replaces the VQ layer by coordinate wise scalar quantization. This drastically reduces the number of hyper-parameters and the codebook size, while still maintaining the reconstruction quality. Quantization is done by first passing it through a bounding function such as $\mathrm{tanh}$ which bounds the dynamic range of the entries to be quantized. This is followed by quantizing the $i^{\mathrm{th}}$ coordinate of the latent representation to one out of $L_i$ values. Here, $i$ varies from $1$ to $d$, where $d$ is the number of channels. Numerical evaluations validate this claim.

Please correct me if I am mistaken in my understanding of the contributions of the paper. I am more than happy to rectify my review if that's the case.

**Strengths:**

VQ-VAE has been successful in obtaining compact latent representations. However, the VQ layer has some issues such as code under-utilization as mentioned in the summary. This paper proposes a simple strategy that imposes the cubic lattice structure while obtaining the quantization codebook. This leads to much fewer parameters to store.

The simplicity of the proposed approach is quite commendable. It is surprising that this hasn't already been studied before. Even though quantization is a non-linear operation, the VAE network can be trained via back-propagation using the stop gradient estimator.

**Weaknesses:**

The idea is well described in the paper. My only concern is the novelty of the contribution of the proposed approach. It is surprisingly simple, which raises my question as to what are the limitations of FSQ, and if it has not been explored before, especially since VAEs have been around for a while.

Can we have any guarantees as to why FSQ performs as good as VQ, even after imposing strict structural constraints? Is there anything particularly special about the tasks considered in the numerical simulations, or can we expect FSQ to perform at par for any application of VAE?

**Questions:**

I have no critical questions right now, expect for the following.

How does the choice of the squeezing / bounding function, i.e., $\mathrm{tanh}$ affect the performance?

I'd be happy to edit my review post the discussion period.

---

> ### Author Response · Authors · 2023-11-14
>
> Thanks for the review, and thorough summary, your understanding of the method is correct.
>
> The main concern of yours seems to be “if the method is so simple and works so well, why has it not been studied before”? As we wrote in the general reply, we think that this is because a crucial insight was missing: "Our key insight is to consider the implicit product codebook created by scalar quantization if you bound each channel, which allows training large generative transformers like on VQ representations (i.e., by bounding each of the $d$ channels to $L$ levels, we get a product codebook of $L^d$ codewords)." We updated the Intro with a clarifying sentence.
>
> Regarding limitations: We believe FSQ-VAEs can work as good or better as VQ-VAEs in any task involving a VAE, and we showed 4 different vision tasks in the paper (generation, colorization, depth estimation, semantic segmentation). As we note at the start of Sec. 3, intuitively, FSQ works well here because the generality of a learned codebook is “absorbed” into the encoder transform, which can morph the data so that it is well quantized by FSQ.
>
> Regarding tanh: We explored using sine instead of tanh but found it to perform worse. There might however be future work that shows a better bounding function than tanh.

---

> > ### Comment · Reviewer_Hp3C · 2023-11-23
> > **Acknowledgment**
> >
> > Thank you for your response.
> >
> > I do believe this is a good paper and should be accepted.

---

### Author Response · Authors · 2023-11-14

Thanks everyone for their reviews. It appears that all reviewers like the simplicity of our proposed approach (“simplicity of the proposed approach quite commendable”, “effective and easy to implement”, “method is simpler (appears to be much simpler than the VQ-VAE cumbersome training and optimization)”, “straightforward solution”). They think that "The experiments are extensive. The authors demonstrate the efficiency of the proposed method under various settings", that there are "sufficient and concrete experiments demonstrate the effectiveness of the proposed method" and that our "method reaches good performance.",

However, some reviewers disagree on the novelty of the approach. We’d like to emphasize that prior to this paper, no one was thinking of using finite scalar quantization (FSQ) as a drop-in for VQ in generative image models or for dense vision tasks. Our key insight is to consider the implicit product codebook created by scalar quantization if you bound each channel, which allows training large generative transformers like on VQ representations (i.e., by bounding each of the $d$ channels to $L$ levels, we get a product codebook of $L^d$ codewords). While it might seem obvious after reading our paper, a priori, it is very surprising that FSQ obtains such competitive results compared to VQ -- despite not using any of its auxiliary losses and tricks.

We strongly believe that the ICLR audience will receive this paper well. Prior to this work, people either used VQ (eg, in the generation lit), or they used scalar quantization without considering the product codebook (eg, in the compression lit).

---

### Public Comment · ~Jiacheng_You1 · 2023-11-30
**According to the code, L=2 can produce significantly unbalanced quantization**

The authors said "Intuitively, we think this is because of the straight through estimation (STE) that we use: for this estimation, we approximate the gradient of the round operation with the gradient of the identity function. This approximation is very bad for L=2, when the quantizer becomes a step function, but gets better and better as we increase L, with diminishing returns as L further increases."

However I found that `L=2` can produce significantly unbalanced quantization using the code in Appendix A.1.
```python
def bound(self, z)
  eps = 1e-3
  half_l = (self._levels_np - 1) * (1 - eps) / 2
  offset = jnp.where(self._levels_np % 2 == 1, 0.0, 0.5)
  shift = jnp.tan(offset / half_l)
  return jnp.tanh(z + shift) * half_l - offset
```
For `L=2`, we have `half_l=0.5` (approximately) and `offset=0.5`, thus `shift=tan(1)` (approximately) and `bound(z) = tanh(z + tan(1)) * 0.5 - 0.5`.

Then `round(bound(z))` will produce `-1` for `z < -1.557` and `0` for `z > -1.557`, see https://www.desmos.com/calculator/jx2p8h4zwp?lang=en-US.

If `z`'s elements can be approximate by $\mathcal{N}(0, 1)$, the quantizer will produce ~6% `-1` and 94% `0`, which is significantly unbalanced.

---

### Meta-Review · Area_Chair_Xdo1 · 2023-12-15

**Metareview:**

This paper proposes to replace the quantization layer of vector quantized variational autoencoders with a scalar quantization strategy. Four experts reviewed the paper and found the work valuable in general. One reviewer has some reservation about the novelty of the work and the simplicity of the proposed approach. The authors have satisfactorily answered most of the raised concerns, such as a lack of analysis on speedup and compression rate.

**Justification For Why Not Higher Score:**

I reached this decision by evaluating the contributions and novelty of the work, taking into consideration both the reviews and the responses from the authors.

**Justification For Why Not Lower Score:**

I reached this decision by evaluating the contributions and novelty of the work, taking into consideration both the reviews and the responses from the authors.

---

### Decision · Program_Chairs · 2024-01-16

Accept (poster)